# A Proposal to Use Determinants of Annoyance in Wind Farm Planning and Management

**Frits (G. P.) van den Berg**

Mundonovo Sound Research, 9953PH Baflo, The Netherlands; fvdberg@mundonovo.nl

**Abstract:** Wind energy in Europe is expected to grow at a steady, high pace, but opposition from residents to local wind farm plans is one of the obstacles to further growth. A large body of evidence shows that local populations want to be involved and respected for their concerns, but in practice, this is a complex process that cannot be solved with simple measures, such as financial compensation. The visual presence and the acoustic impact of a wind farm is an important concern for residents. Generally, environmental noise management aims to reduce the exposure of the population, usually based on acoustics and restricted to a limited number of sources (such as transportation or industry) and sound descriptors (such as $L_{den}$). Individual perceptions are taken into account only at an aggregate, statistical level (such as percentage of exposed, annoyed or sleep-disturbed persons in the population). Individual perceptions and reactions to sound vary in intensity and over different dimensions (such as pleasure/fear or distraction). Sound level is a predictor of the perceived health effects of sound, but explains only part of the reaction. The positive or negative perception of and attitude to the source of the sound is a better predictor of its effects. This article aims to show how the two perspectives (based on acoustics and on perception) can lead to a combined approach in the management of a wind farm aimed to reduce annoyance, not only on a sound level. An important aspect in this approach is what the sound means to people, leading to the following questions: is it associated with the experience of having no say in plans, does it lead to anxiety or worry and is it appropriate? The available knowledge will be applied to wind farm management, including planning as well as operation.

**Keywords:** wind farm; annoyance; social acceptance; worry; noise sensitivity; noise management

## 1. Introduction

In 2021, wind energy installations in Europe (EU and UK) had a total capacity of 236 GW [1]. This corresponded to 15% of the electricity consumed in 2020, where electricity consumption was about 23% of total energy consumption in Europe [2]. This was expected to grow in the next few years with 15 to 20 GW per year, but in 2021, the EU set a new target of 30 GW per year to arrive at a total of 451 GW in 2030 [3]. Achievement of this will depend on how national and local authorities in Europe are able to balance the need for renewable (wind) energy with the growing opposition to wind farms.

Renewables Now [4], the 'global renewable energy community of actors from science, governments, NGOs and industry', reported substantial growth in renewable energy, but observed the share of 'modern renewables' (wind, solar, biofuel) in the total energy consumption to be less impressive. In the years 2009–2019, the total demand for modern renewables increased to 15.1 EJ (exajoule), but total final energy consumption grew to 60.9 EJ. Thus, renewables could not even keep up with the increase in energy demand during this decade. The report noted that, with respect to wind energy, there are policy and social factors that slow progress, such as delays in permitting procedures and public opposition, and the shift to auctions of wind projects, which has an indirect negative impact on social support for wind farms. The wind industry in Europe acknowledged in a 2022 report that governments' commitments are still 'fairly modest' and expected that in a

'realistic scenario', Europe will install at an average rate of 17.6 GW a year, well below the 32 GW per year necessary to reach the 40% renewable energy target [1]. This is even more unfavourable in a 'low scenario' of minimal policy support, where governments do not solve restrictions in permitting issues or spatial planning requirements. Lack of such policy support is also a barrier to renewable energy deployment, as reported by the International Energy Agency (IEA) [5]. The other barriers are grid integration and a lack of grid availability, financial risks and lack of remuneration, as well as 'challenges' concerning social acceptance of wind (and hydropower) projects that increasingly cause delays or cancellation of projects. The EU Joint Research Centre observed that 'social acceptance is a key challenge for the deployment of wind energy and could limit the overall wind resource we are able to exploit to meet climate change targets' [6]. From a survey among experts involved in wind plans in 13 EU countries, Dütschke et al. concluded that, similar to an earlier study [7], 40% of the plans were met with resistance, which led to a change in plan, delay or termination of the plan [8]. Social acceptance rests on a network of factors that interact, so there is no simple way to include it in wind farm planning; it "cannot be achieved through isolated 'fixes' such as community benefits or just more consultation, but must be a more fundamental change" [6]. Hirsch and Sovacool [9] recalled Langdon Winner who, in 1982, noted "an obsession with economic indicators and cost-benefit analyses among policymakers", which "obscures historical and social concerns that influence energy policy", with the result that "energy policy remains short-sighted about noneconomic considerations" [10].

Thus, it can be concluded that in Europe, social acceptance is a major barrier for the realization of renewable energy goals. In this paper, it will be shown that the actual impact of and health concerns related to a wind farm are important determinants of the opposition to wind farms. To realize the ambitious renewable energy goals, more attention should be paid to what drives these concerns and to ways to address them. The primary aim of this paper is to include these concerns in the planning and management of wind farms, based on the general knowledge of effects of noise on residents and the more specific knowledge concerning wind turbines.

Section 2 covers the many acoustic and non-acoustic factors that influence the relation between annoyance and noise level, especially at an individual level. How do we interpret sounds, how do sounds become associated to meaning? A host of studies and reviews show that for environmental noise sources, there is often a significant relation between its noise level at a residence and the annoyance that residents feel. The added annoyance due to source and noise characteristics can be taken into account by using different relations per noise source and corrections based on noise character; but even then, the noise level only explains a small part of the total annoyance. Worry about possible health effects, possibly caused by chronic annoyance, is an important determinant of (a lack of) social acceptance. The view developed in Section 3 will be based on general knowledge of (psycho)acoustics and health effects from noise (including annoyance), and supported by the more limited knowledge related to wind turbines. It will focus on factors that can be influenced on a local level, independent of national policies or regulations. The purpose of this is to arrive at an understanding of how to decrease the impact of a wind farm and help in building trust that a windfarm can and must be an acceptable 'neighbor'.

## 2. Perception and Effects of Sound

### 2.1. Meaning of Sounds

Environmental sound usually refers to sound from transport sources and business activities, but can extend to all sounds in the (home) environment. In practice, environmental sound levels extend from about 20 dB(A) to less than 80 dB(A). At both ends of this range, there is generally agreement about the perceived quality, either very quiet or very noisy, respectively. However, in between, there is a broad range where some find the situation noisy and others do not. This is illustrated in Figure 1, where respondents near Dutch wind farms at any specific sound level in the range 30 to >45 dB $L_{den}$ (average

A-weighted day-evening-night sound level) gave qualifications ranging from 'do not hear' to 'very annoying' [11]. The acoustic quality of the sound and a host of other non-acoustic factors are at the base of these differences.

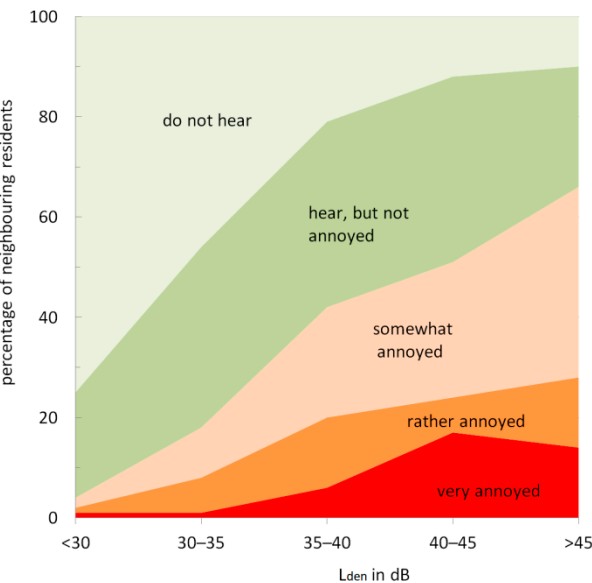

**Figure 1.** Reactions to wind turbine noise.

In psychological terms, sound can have different affective (i.e., related to emotion) dimensions, which are as follows: arousing, exciting, pleasant, relaxing, sleepy, gloomy, unpleasant, distressing [12]. Annoying, the only affective aspect used in environmental assessments, is in between exciting and distressing and the opposite of relaxing rather than pleasant [13]. Sound is evaluated at the following levels: as an immediate, unconsciously processed impression; its information content; its relation to the context present; its loudness [14]. Playing children 'belong' in many home environments, but not at night or in an industrial environment. In addition, the sound of playing children has an influence on road traffic sound; the total sound is experienced as "better" and "more natural", but not "calmer" than the road traffic itself [15]. At the same physical sound levels, pleasant sounds are experienced as less loud than unpleasant sounds [16]. The affective meaning of wind power was studied in a Swiss survey [17], where respondents were asked to give up to five associations they had with wind energy and then rate this according to their affective value, ranging from 1 (very negative) to 7 (very positive). Positive associations were predominantly related to renewable wind power production and categorized in the study as ecology, power production and wind. Dominant negative associations were related to environmental impacts, including landscape, noise and wildlife. Older respondents and respondents not familiar with wind energy had significantly fewer positive associations compared to younger persons and those familiar with wind energy. When comparing groups more or less supportive of wind energy, all groups had positive and negative associations, but the balance shifted from more positive to more negative when supportiveness decreased. Zaunbrecher et al. found similar types of spontaneous associations with wind farms, including visual impact, environmental effects and energy (and related terms) [18]. Hirsch and Sovacool state that wind turbines have great symbolic value and attractive to some, but repulsive to other stakeholders. In a positive sense, they are popular symbols of 'progress, modernity and environmental consciousness'; in a negative sense, they represent the industrialization of relatively unspoiled and natural environments [9].

### 2.2. Perception

People are very different from sound meters; we do not hear sound as an acoustic presence, but as a representation of a source. When we hear a sound, we cannot but

associate it with a source. There is no 'acoustic' impression or observation, but a voice that speaks or a dog that barks, a car or a bicycle bell; we hear what causes the sound [19]. If consciously heard, this leads to associations, including the pleasure of a familiar voice, the fear of damage from a billowing storm or the expected sleep disturbance from the nearby wind farm. The sound as such is not the cause of stress or fear, but the awareness of the source of sound is. Thus, a wind turbine is not perceived as a neutral object, a vertical bar with three 'spokes' on top, but as a human-built machine that may summon positive or negative associations.

People see, hear, feel, smell—in short, people experience all aspects of an environment at the same time with each person's personal associations. Coherence between different aspects of perceptions is important for the total sensation. In environmental research, there is no indicator for the total perception of an object in the environment (such as a road, airport or wind farm) nor for its total impact, but only for specific impacts, separately for each object (source) and sense (visual, aural, odour, tactile). Interactions between these impacts is not taken into account. For example, a visually highly regarded natural environment is less appreciated when unnatural, mechanical sounds are heard [15]. In a nationwide survey, Van Rompaey et al. [20] used pictures of various landscapes with or without wind turbines. A visual quality index was based on the percentage of forest, built up area, hilliness/flatness, and absence/presence of human elements. The survey results showed that wind turbines affected the visual quality; low-quality landscapes were perceived as improved and high-quality landscapes as degraded. According to Frantal et al. [21], the influence of wind turbines on visual aspects of the landscape is highly dependent on the local environmental and socioeconomic context. In addition, annoyance from the visual presence of wind turbines may add to and even reinforce the noise-related annoyance (and vice versa). Voicescu et al. found that flicker shadow annoyance was best predicted by noise level [22], whereas Haac et al. found that visual impression (did respondents like the look of the wind farm?) was the most important factor to predict noise annoyance (the next important factor was noise sensitivity) [23]. Michaud et al. attempted to integrate separate annoyances from a wind farm into 'aggregate annoyance' and higher scores on this scale showed to be correlated with a number of health conditions (blood pressure, perceived stress, sleep quality, physical health, psychological well-being, chronic pain, tinnitus, migraines/headaches and dizziness) [24].

### 2.3. Noise Annoyance and Other Health Effects

It is generally assumed there is a monotonous increase in the percentage of annoyed persons when the noise level increases. How closely changes in percentage follow changes in noise level is usually expressed by means of a correlation coefficient (c.c.). This is a number (from −1 to 1) that shows how strong the relationship is between two variables, such as a c.c. of 1 or −1 means that changes in one variable are perfectly predicted by changes in the other variable. Closer to zero (0) means there is no perfect relation, with 0 indicating no relation at all. It has been shown repeatedly that noise level explains only part of individual annoyances. In a large number of studies, Job [25] found relations between sound level and individual annoyance with correlation coefficients (c.c.), varying from 0.22 to 0.61, with a mean value of 0.42. This means that the variance (square of c.c.) was 0.04 to 0.37, with an average of 0.17 or 17%. This 17% is the degree in which a variation in noise annoyance is explained by a variation in noise level. Almost 30 years later, Brink [26] reported similar values; generally, variances were from 0.1 to 0.2. The annoyance at the group level (groups with many individuals each) was better correlated to noise level, with a variance of 0.31 to 0.98 and an average of 0.67. For wind turbine sound, correlation coefficients were found in the same range. In a Dutch survey, Pedersen et al. [27] found a c.c. between wind turbine sound level and annoyance of 0.51. In an earlier Swedish study [28], the c.c. was 0.42.

In the context of the WHO noise guidelines, Basner and McGuire [29] concluded that transportation (air, road, rail traffic) noise, in the range of 40–65 dB $L_{night}$ (average

A-weighted night-time sound level), had a direct effect on sleep, as it influenced objectively measured physiological indicators of sleep. However, they also concluded that sound level may not directly influence self-reported sleep disturbance, but indirectly through individual factors, demonstrated by the following statement: "This suggests that for self-reported measures it is annoyance or attitude to the night-time noise that may be driving the increase of reported sleep disturbance outcomes with $L_{night}$ level" [29]. Van den Berg et al. [30] found similar results from a survey that investigated noise annoyance and self-reported sleep disturbance from various noise sources (transportation, neighbors, industry); correlation coefficients between both effects varied from 0.75 to 0.84 and the annoyance, thus, explained 56–70% of the variation in sleep disturbance. This is high in comparison to noise level, which explains, on average, only 10% of self-reported sleep disturbance [31]. Meaningful noise may disturb sleep more than meaningless or neutral noise [30,32]. For example, 'meaning' could be a perception of inconsiderateness or even malignancy at the part of the noise source, and thus may be related to the attitude towards the source. It has been shown for wind turbines and church bells that attitude can influence the effect of noise [32,33]. Michaud et al. found that self-reported sleep disturbance was not related to wind turbine (WT) sound level, but closing the window because of sleep disturbance had a strong association with annoyance [34].

Cardiovascular health effects, such as ischaemic heart disease and hypertension, have been found to correlate with relatively high sound levels [35], not with the lower levels of WT sound. Bräuner et al. [36] found "suggestive evidence" of an association between long-term exposure to WT sound and atrial fibrillation (AF) amongst female nurses. They also found that the relative risk of AF in the highest exposure group was comparable to the relative risk in the large non-exposed group [37]. Chaban et al. [38] asserted that WT infrasound could have a direct effect on the heart, based on laboratory tests on live heart tissue. However, this assertion was questioned because the actual level of WT infrasound and the effect of (infra)sound on a human body were based on erroneous assumptions and an inappropriate setup of the experiment [39,40]. Although a direct effect of sound cannot be ruled out, it is plausible that cardiovascular effects are the result of noise-induced stress [41]. Chronic annoyance leading to stress is probably the most important route leading to further health consequences [42]. This may explain why Poulsen et al. did find a relation between WT sound level and the use of sleeping medication and antidepressants [43], but did not find a relation between WT sound level and hypertension [44] or conclusive evidence of an association between WT sound level and a heart attack or stroke [45] (similar to Bräuner et al. [46]). Thus, for wind turbines, there is no clear evidence that the sound level is directly related to sleep disturbance, stress or other health effects, but there is a correlation between such effects and annoyance [47,48]. Michaud et al. [24] have shown that the sum of annoyances from noise, blinking lights, flicker shadow, visual impact and (to a lesser extent) vibrations are correlated to perceived health symptoms.

*2.4. Factors Influencing Noise Annoyance*

Miedema et al. [49] collected survey responses from about 64,000 respondents to determine the relation between sound level and annoyance, with respect to transportation sound. They also investigated factors that could influence this relation [50], using data from almost 43,000 respondents in 34 studies. Eight of these factors were demographic (gender, age, level of education, occupation, household size, homeownership, dependency on and use of noise source), two were 'attitudinal variables' related to personal traits (fear of noise source and noise sensitivity). The response for each variable was assigned to a number of categories, e.g., four categories of education level (1st level to university level) or three categories of noise sensitivity (tertiles of 0–100 scale). The results showed that noise annoyance was not related to gender, but significantly related to the other demographic factors, albeit to a low or very low degree. In nearly all cases, the extra noise annoyance related to each factor was equivalent to a change in day–night level DNL (=Ldn $\approx$ Lden) of

−2 to +2 dB. Larger effects were found for young (10–20 years) and old persons (70+) and those economically dependent on the noise source in question; these groups were relatively less annoyed (equivalent to 4, 3 and 2.6 dB, respectively). In contrast, the personal factors noise sensitivity and fear had a far larger influence on noise annoyance. Noise sensitivity was assessed with a single question ('Are you sensitive to noise?"); the extra annoyance for those highly sensitive (upper tertile) was equivalent to 11.2 dB, when compared to the low sensitive persons (lower tertile). In the reviewed studies, fear was assessed with different questions, but all were related to aircraft/road/rail accidents. Fear had an even stronger effect; the extra annoyance in the highest tertile was equivalent to 19.5 dB when compared to the lowest tertile. Miedema et al. suggested that this effect could be because of a (personal) disposition towards fear or an actual experience of fear [50].

The relation between annoyance and worry (one of the expressions of fear) has been investigated by Van den Berg et al. [51], who used results from an environmental health survey in the Amsterdam area. Respondents were asked if they were annoyed by a number of noise and odour sources and if they were worried about possible hazards. Answers to these questions were given on an 11-point scale, ranging from 0 (not annoyed/worried at all) to 10 (extremely annoyed/worried). The list of nine noise sources, as well as the list of five odour sources, included road traffic and aircraft. The list of thirteen possible hazards included situations related to road traffic and aircraft, such as living in a busy street, living close to an airport and living under the air route of a main airport. The coefficient for the correlation between annoyance scores from road traffic and worry scores of those living near a busy road was 0.49 and for odour annoyance and living near a busy road, it was 0.46. For aircraft noise annoyance and living near an airport or near an air route, the correlation coefficient was higher, 0.63 and 0.59, respectively. Associations between aircraft odour annoyance and living near an airport or near an air route had similar correlation coefficients (0.57/0.56). On average, the annoyance score increased with two units for every unit increase in worry score; this applied to noise as well as odour annoyance. Further results suggested that being worried seemed to be related to a disposition rather than an actual threat, as worry scores for very different situations were also significantly correlated. For the 78 possible pairs of worry scores, each correlation coefficient was ≥0.45 and in half of the cases, it was even ≥0.60. This may be obvious for related situations (e.g., living near an airport and living near an air route), but less so for unrelated situations (e.g., living near an airport and living below sea level). However, results showed that correlation coefficients for unrelated pairs were not clearly lower than for related pairs.

The large contribution of worry to annoyance from wind turbine noise has been confirmed in the Canadian Noise and Health Study [34], where the contribution to noise annoyance of all personal and situational variables included in the survey was studied. Eleven variables explained most of the noise annoyance. Of these, six variables concerned a response to wind turbines (closing bedroom window due to wind turbines, annoyance from blinking lights/vibration/sight, self-reported sleep disturbance, complaints about wind turbines), together leading to a variance of 0.46. This means that all these reactions to the wind farm operation explained 46% of the annoyance from noise. The other factors, including sound level, explained a further 14%. When the variables were restricted to those that were expected not to be a direct response to wind farm operations, the resulting eight variables explained 40% of the noise reaction. Here, the single most important factor was concern about physical safety (explaining 17%), followed by wind turbine sound level (11%) and noise sensitivity (7%). From his review, Simos concludes that when wind turbines are built, people in their neighborhood may experience anxiety and distress and anxiety is an important risk factor linked to cardiovascular diseases and mental health conditions [47].

### 2.5. Health Concerns Related to Wind Farms

The impact on health is one of the main topics in the debate on effects of wind farms, which is reflected in many reviews (e.g., [37,47,52–54]). Van Kamp and van den Berg concluded that health effects on residents near wind farms can be related to exposure to and/or

annoyance from the wind turbines, but they are also, and perhaps more so-, the result of other factors, such as participation in the planning process, procedural justice, feelings of fairness and balance of costs and benefits from wind turbines [37]. Simos et al. conclude from their review that when wind turbines are erected, persons in their neighborhood may experience anxiety and distress, although annoyance is the only symptom backed up by solid scientific evidence [47]. This lack of evidence for health effects directly related to wind farm exposure does not mean that residents are not concerned. Chapman et al. [55] suggested that in Australia, public concern increased when anti wind farm groups started to stress health problems. Stirring up such 'health anxiety' in the planning phase could lead to interpreting common health problems as being caused by wind turbines [56]. In Canada Deignan et al. analyzed newspaper coverings of wind farms and health in different geographical areas and found that four 'fright factors' were frequently mentioned [56]. Nearly all articles (94%) mentioned 'dread' (as such or in other terms) and about half of them mentioned 'poorly understood by science', 'inequitable distribution' and 'inescapable exposure'. Such terms may lead residents to worry about the effect that a wind farm may have for them personally. Dällenbach and Wüstenhagen used data from a national Swiss survey on the acceptance of low-carbon technologies and policies and from a public consultation process for a specific wind farm plan [57]. They found that within 1 km from the planned wind farm, noise concerns were related to the expected noise impact. There was no such relation for 93% of the objectors who lived at larger distances, where calculated noise impacts were low or negligible. A similar result was found from the survey; 18% of the respondents within a 5 km radius from a planned wind farm expressed noise concerns, as did 19% of the respondents living at greater distances. Of the respondents familiar with wind turbines (they could observe existing wind turbines) 12% expressed noise concerns, which differed significantly from the 20% of respondents unfamiliar with wind turbines. The authors concluded that "(lack of) familiarity with wind energy and issues related to procedural and distributional justice appear to provide a better explanation for the geographical spread of noise concerns". Cousse et al. found that the first association of Swiss residents with wind energy is often positive, followed by somewhat more negative 'second thoughts' [17]. They suggest that social acceptance could increase when project developers make sure that the local population can overcome such "second thoughts".

In a survey amongst German residents, Zaunbrecher et al. investigated preferences in two situations that were presented to the respondents [18]. One was a plan for a new power line in their neighborhood, the other for a new wind farm. Characteristic 'attributes' in the scenarios were as follows: health complaints, compensation payment, location (in forest, near other infrastructure, on a field) and distance to mast (pylon or wind turbine). The results show that health concerns were the most important attribute, while compensation payments were the least important. This was the same for both pylons and wind turbines and also for mobile phone base stations from another study.

Crichton et al. [58] studied anticipated health concerns by exposing a student population to moderate wind turbine sound with or without added infrasound, after either (1) a TV presentation of residents, attributing symptoms to a local wind farm or (2) TV interviews of experts, stating wind farm infrasound could not cause symptoms. Participants that viewed the first presentation reported significantly more and more intense symptoms in both exposure groups (with or without infrasound), when compared to the second presentation. As in this study, the infrasound exposure was very low (40 dB at 5 Hz); Tonin et al. [59] reproduced the study with realistic wind turbine infrasound of an 80 dB average level (91 dB peak level), administered by way of headphones (all sound levels unweighted). The participants in this study, from a large age range and professional backgrounds, were divided in four groups according to the two exposure conditions and two presentation types (that were now viewed before exposure). The authors concluded from this study that the infrasound itself did not influence the intensity of symptoms. The expectancy manipulation did have some influence, but not to a significant degree. However, the notion of harmfulness of infrasound that participants already had before the study did have a

significant influence. As mentioned by Crichton et al. [58], it is not only for wind turbine sound that expectations influence the number and intensity of health symptoms, but it has been found also for other environmental agents, such as pesticides or odour.

Of course, the lack of evidence for a direct relation between the impact of a wind farm and reported health effects do not exclude an indirect relation or a relation obscured by other factors. Several studies show that there is a relation between a subjective perception (annoyance) and self-reported health effects (e.g., see [34,60,61]). The lack of a direct relation may be due to other factors, such as sleeping with open/closed windows, side of the bedroom façade or noise sensitivity.

Noise seems to be the dominant residential impact of wind turbines (e.g., see [47,62,63]) and concerns about its effects remain a dominant reason to oppose wind farms. Perceived fairness and visibility are certainly also key issues, but are less easily identified and objectified. In 1989, Wolsink found that in the case of three Dutch wind farms and one single, large (at that time) 1 MW wind turbine, visual impact was the main source of opposition, but people "know that using visual criteria is probably not the best bet in formal procedures" [64]. As Hirsch and Sovacool put it, " . . . (opposition) . . . will often focus on more 'rational' reasons to make cases to regulatory bodies and others. After all, an environmental agency would respond more favourably to opponents' assertion that endangered species of birds would be killed by wind turbines than to claims that the technology seems 'out of place'" [9]. In fact, some visual factors are highly correlated to sound level [22,23] and the combined annoyances of visual factors and noise explain most of the variation in the perceived health effects [24].

Fear is an important driver of animal and human behavior. According to the Merrriam-Webster dictionary, fear is an "unpleasant, often strong emotion caused by anticipation or awareness of danger [65]". Feelings of fear or worry (which is an anticipation of fear) are not necessarily based on actual or 'objective' danger, such as that acknowledged by institutions or authorities, but can be based on individual perceptions, beliefs and/or cognitive or attitudinal dispositions.

## 3. Application to Wind Farm Management

### 3.1. Social Acceptance

Even though a positive attitude towards wind energy is linked to a higher acceptability on an individual level, several authors have stressed that a positive attitude towards wind energy in general does not imply a higher degree of acceptability in a local community [66,67]. Wolsink concluded from data gathered between 1986 and 2002 that planners assume support because of the environmental benefits, but disregard feelings about equity and fairness [66]. A recent study of two large Finnish wind farms showed that residents were often passive with respect to participation, but active participation was perceived as symbolic and did not lead to a real influence of the participants [67]. Dütschke et al. concluded from their survey under experts in the wind sector that project developers start some form of public participation mainly after preparation of the project, and when the project officially starts, public opposition may already have formed [8].

Many studies stress the importance of a fair planning process and of local involvement and participation. For example, in surveys in Poland and Germany, Liebe et al. found that respondents were willing to accept new wind turbines in their neighborhood if they could participate in decision making, if turbines were owned by a group of citizens, and if the generated electricity was used in the region (and not exported) [68]. The authors also concluded that, as respondents who already have turbines in their area were more inclined to accept new ones (compared to those not familiar with them), negative opinions may be overestimated in the planning and implementation process. In a case study of a Scottish wind farm, MacDonald et al. interviewed 'key players' from different stakeholders [69]. Inadequate engagement of the community led to negative perceptions of the plan and a non-negotiated community benefit program increased these negative perceptions, as this program was perceived as ineffective and a means for the developer to obtain planning

permission. With a survey amongst a large sample of German residents, Gölz found that variations in perceived fairness explained variations in acceptance to a large degree, where fairness was influenced mainly by trust in local stakeholders and attitudes towards wind power [70]. However, even in the four regions of Germany (states in the north, east, south and west), they found that the influences of these and other factors could vary considerably. This led to the conclusion that acceptance will profit from more respect to regional views and conditions. The authors noted a possible conflict with the German practice of public tendering of wind farms. Wolsink and Breukers also found differences between regions (Netherlands, North-Rhine Westphalia and England) that were rooted in different historical developments and planning approaches [71]. They concluded that a strategy that included facilitating local ownership, early participation in project planning and development and favouring initiatives in socio-technical innovation enhances legitimacy of and reduces opposition to projects. These differences illustrate that social participation is advisable for each project, as only then local issues will come to light.

Potential problems with the acceptability of new projects are not unique for wind energy. For example, in the case of a municipal waste incineration plant, Subiza-Pérez et al. found that predictors of acceptance were trust in the information provided by the local government and perceived risk for human health [72]. Cataldi stated that for geothermal energy projects, the main factors contributing to social acceptance are environmental impact, adverse effects on human health and tangible benefits for the local population [73]. With respect to the expansion of two European airports, Liebe et al. concluded from a survey that aspects of environmental and participatory justice have much more influence on acceptability than economic and distributive justice (a more even spread of impacts) [74].

The acceptability of a wind farm is not about accepting renewable or wind energy as such, but about involving, and not only informing, the community in decisions on the planning and operation of a wind farm. Simos et al. state that "The empowerment of affected populations is a central tenet of health promotion. Its efficacy and usefulness have been demonstrated in many settings across the world. Empowerment also tends to increase the acceptability of projects" [47]. In their review, Simos et al. recommend for people to recognize the feelings of local residents and to take their points of view into account. In practice, however, if there is participation, it is often unclear which factors actually are open to participation and if the community has the power to influence decisions. Financial compensation in any form (house price compensation, community benefit schemes) can be part of an arrangement, but is not a sufficient instrument to achieve acceptance and can even be viewed by some as a bribe, a way to 'buy acceptance'. According to Zaunbrecher et al., health concerns were the most important and compensation payments the least important issues for residents [18]. From a recent survey that involved residents living near four Dutch wind farms, it was concluded that financial participation did not rectify the insufficient involvement of residents in the planning process [75].

In the proposal below, the focus is on the involvement of the community, predominantly with respect to the impacts they expect to experience in the planning phase and will actually experience in the operational phase. All measures that lead to the prevention or reduction in annoyance will also lead to a reduction in perceived health effects related to annoyance. The proposed approach can also be applied to existing wind farms.

### 3.2. General Considerations

From the previous sections, the following conclusions can be drawn:

- Noise annoyance is the primary reaction to noise (i.e., sound that disturbs, unwanted sound).
- Further health effects are likely to result from chronic noise annoyance, although a direct physiological effect of sound is not excluded.
- Annoyance from a noise source is related to the sound level from that source, but only to a modest degree; other factors contribute substantially to noise annoyance.

- Of these other factors, fear, worry or concern in relation to the noise source and noise sensitivity appear to be the most important factors. Because of beliefs, associations and experiences with the sound source and possibly the characteristics of the sound, it may carry a meaning that the presence of the sound source is 'unsafe'.
- Noise is only one of many possible environmental factors and its impact is not experienced separately from other factors.

From this, it can be inferred that reduction in the impact of environmental sound can be reached in different ways. One way is to reduce the impact of sound is either by reducing sound level or by reducing annoying characteristics of the sound (such as tonal content) or other exposures from the same source. The usual approach is to set a sound level limit to prevent residents from being exposed to unacceptable high sound levels. A limit is usually based on an acceptable level of annoyance and may be derived from sources with more or less similar effects or exposures. A procedure to derive a limit by comparison with other sources is described by Fredianelli et al. [76]. Others, such as the WHO [48] and Davy et al. [77], have proposed limits based on an acceptable prevalence of annoyance. In these cases, the character of sound can be included or implied in the limit, but there is no bonus for an operator who tries to minimizee annoying characteristics. In addition, a certain prevalence of annoyance (and possible other effects) is taken for granted and other ways to reduce annoyance are not taken into account.

A second way to reduce the perceived impact of a wind farm is to reduce negative perceptions of and/or associations with the noise source. This, of course, is neither simple nor straightforward, as the perceptions and associations may, at least in part, be realistic and reasonable. However, they can also have another cause; amongst others, earlier experiences or 'fear of the unknown' or a lack of control can play a role. To explore this way of reducing annoyance, it is also necessary to consult members of the exposed population, as any mitigation measures should diminish worry in that population. The third important factor, noise sensitivity, cannot be reduced by itself, as it is a stable, individual characteristic. However, a reduction in sound level and noticeable characteristics will help. Apart from that, awareness of noise sensitivity may be of use in noisy areas; if the sound level cannot be reduced, it may help to warn people who consider moving there when they think of themselves as being noise sensitive. For those living in the area, individual mitigation measures may help.

### 3.3. Wind Farm Impact

The general conclusions in the previous section also apply to the case of wind farms. In a recent review of the residential effects of wind turbine noise, Van Kamp and Van den Berg [37] concluded that health complaints (other than annoyance) related to wind farms are primarily related to annoyance, which again depends on a range of non-acoustic factors and the actual exposure. Such factors include "noise sensitivity, attitudes towards wind turbines, health concerns, visual aspects and aspects related to the procedure preceding the building of a wind park. The role of factors such as participation in the planning process, procedural justice, feelings of fairness and balance of costs and benefits from wind turbines are even more strongly supported by current evidence" [37]. It is not just the sound from a wind farm, but also visual effects and perhaps vibrations that can lead to annoyance. Simos et al. [47] suggest that a balance between energy production and healthy living conditions can be found by reducing the impact by turning wind turbines on or off or changing their rotational speed, depending on conditions such as time of day or weather.

Citizens confronted with a wind farm plan in their area may be concerned about the consequences for their health and other environmental effects (such as animal mortality). Not taking these concerns seriously is likely to fuel more opposition. At least part of the concern is not unreasonable, as many national limits for noise and shadow flicker do not exclude the occurrence of (severe) annoyance and possible further health consequences. Thus, some residents will be affected if the wind farm is built and operational, and worrying about this in the planning phase may lay the roots for (more) annoyance in the operational

phase. On the other hand, at least a measure of severe annoyance (and possible further health consequences) is generally considered acceptable when it concerns transport noise. Furthermore, if society does not embrace renewable energy in the short term, there will be other serious public health consequences related to climate change.

The present approach to environmental effects is based on the following 'objective' quantities: sound level, shadow flicker time and safety risks. Usually, these are calculated and it may be impractical, or even impossible, to measure these quantities locally. However, most residents do not care that much about objective limits; they do not want to be affected by the presence of the wind farm, especially not when the pros and cons are out of balance. In the international Wind Turbine Noise Conference 2021, this was explored in two sessions concerning 'being good neighbors' in relation to wind farms. Employees of a wind farm operator said that they want to reduce annoyance from the wind farm as much as possible and residents near a wind farm said they would welcome efforts of their neighboring wind farm operator to reduce the impact on residents. Good neighborship is a form of social recognition, which is one of the key determinants of health [47]. In a UK court case, the Lord Justice stated that "Sticking to the rules is an aspect of good neighborliness but it is far from the whole story-in law as in life" (quoted in [78]).

### 3.4. Proposed Approach: Reduction in Annoyance from Wind Farms

The preceding information suggests that a noise management approach based on reducing annoyance as much as possible seems to be promising. It is also a way to express good neighborliness, which should be pursued in order to achieve sustainable energy goals. Of course, it may not eliminate annoyance entirely, as it seems unlikely that one will be able to find solutions for every individual resident. This approach is, in fact, an application of the ALARA principle (as low as reasonably achievable), where 'reasonable' is an optimal balance between the cost of mitigation and the reduced exposure of the target group. EAN, the European ALARA Network, strives to apply this principle "for the management of occupational and public exposures and patients in all exposure situations" [79]; although, as of yet, it has not been applied to wind energy. General measures would apply to the population near the wind farm and can be based on what is known about residential effects of wind farms. An individual perspective may help to understand residential reactions, certainly with respect to persons who are affected more than average because of their situation or their condition.

Noise annoyance management can be based on measures in the planning phase and in the operational phase. This is part of a process that acknowledges the importance of the factors mentioned above (participation in the planning process, procedural justice, feelings of fairness, balance of costs and benefits, other exposures).

Planning phase:
1. It is important to respect residents' worries in relation to the planned wind farm and to address these in an early stage.
2. One must aim to minimize the impacts, including no (or as little as possible) flicker shadow; wind turbines with below-average sound production; no permanently blinking aircraft warning lights; consideration of visual impact.
3. Perhaps other measures can be discussed, such as synchronization of the wind turbines' rotation to reduce the visual disturbance or possibilities to reduce the rhythmic sound character (amplitude modulation).
4. For the most impacted and/or vulnerable residents, individual mitigation measures are possible, such as planting trees, moving the bedroom to the quiet side, adding insulation, etc.
5. In the planning phase, it is important to install trust that mitigation measures in the operational phase, where necessary, will be implemented and which financial and technical means are available to do this.

Operational phase:

6. As amplitude modulation (AM) of wind turbine sound is an important characteristic, and annoyance is equivalent to an increase in sound level of 3 to 5 dB or more, reduction in this can have a significant effect. Only one case study has been reported that concerned AM mitigation [80].

7. The presence of tonal noise is not a universal characteristic of wind turbines, but also not a rare phenomenon. Tonal noise leads to increased annoyance and mitigation measures should be applied as soon as possible. For the time the tonal noise is present, this should be compensated by a reduction in sound level and/or exposure time.

8. Residents can be consulted about the actual impact of the wind farm, and especially about situations where the annoyance is above average. Consultation is possible with representatives from the community (sounding board group), periodic meetings and/or surveys and more permanent two-way communication means (website, app, complaint desk).

9. It is reasonable that a wind farm operator sets funds aside to implement mitigation measures.

In recent advice to the city of Amsterdam, a small expert committee, including the author, on health effects of wind turbines proposed noise limits and a reduction in the impact of planned wind turbines, in line with the recommendations above [81]. The reduction could apply to sound level, tonal content or the rhythmic character (amplitude modulation) of the WT sound and flicker shadow. Measures could be considered in the planning phase (choice of location and turbine type) and when operational (monitoring concerns and complaints of residents).

## 4. Conclusions

The lack of social acceptance of wind energy plans is likely to slow down the transition to sustainable energy. To be able to achieve the goals the European Union set for 2030, understanding the concerns and interests of residents near projected wind farms and acting to reduce the planned and actual impacts of the wind farm will be helpful. A large number of studies and reviews show which factors determine opposition to the planning and operation of wind farms. Important factors are perceived fairness, respect and trust, factors that are not easily translated into practical terms. However, if it is understood that residents worry about the impact of a wind farm and want to shift the balance of advantages and disadvantages to take their interests and concerns seriously, measures can be taken to address these issues. The recommendations in this paper are based on reducing negative impacts and involving the community, with the aim that a project developer establishes, from the start, a policy of good neighborhood with the resident community, by involving residents in minimizing the negative impacts.

**Funding:** This research received no external funding.

**Institutional Review Board Statement:** Not applicable.

**Informed Consent Statement:** Not applicable.

**Data Availability Statement:** Not applicable.

**Conflicts of Interest:** The author declares no conflict of interest.

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
