# Peer review of "A Proposal to Use Determinants of Annoyance in Wind Farm Planning and Management"

_2674-032X, doi:10.3390/wind2030030_

Round 1

Reviewer 1 Report

Dear author,

First of all I would like to thanlk you for a well written and interesting article and it would certainly be interesting if and when the advice are implemented and evaluated.

I have a few minor comments that I hope you consider beneficial for the overall quality of the paper.

Best wishes/
Reviewer

Detailed comments:

In sec 2.3 I would like the author to include comments and references to the following papers as I think they are relevant to that chapter:

Poulsen A.H., Raaschou-Nielsen O., Peña A., Hahmann A. N., Baastrup Nordsborg R.,  Ketzel M., Brandt J., Sørensen M.,(2018a) Long-term exposure to wind turbine noise and redemption of antihypertensive medication: A nationwide cohort study, Environment International, Volume 121, Part 1, Pages 207-215, ISSN 0160-4120, https://doi.org/10.1016/j.envint.2018.08.054.

Poulsen AH, Raaschou-Nielsen O, Peña A, Hahmann AN, Nordsborg RB, Ketzel M, Brandt J, Sørensen M. Short-term nighttime wind turbine noise and cardiovascular events: A nationwide case-crossover study from Denmark. Environ Int. (2018 b) May;114:160-166. doi: 10.1016/j.envint.2018.02.030.

Poulsen, A. H., Raaschou-Nielsen, O., Peña, A., Hahmann, A. N., Nordsborg, R. B., Ketzel, M., Brandt, J., & Sørensen, M. (2019). Impact of Long-Term Exposure to Wind Turbine Noise on Redemption of Sleep Medication and Antidepressants: A Nationwide Cohort Study. Environmental health perspectives, 127(3), 37005. https://doi.org/10.1289/EHP3909

Line:
-219 missnig reference (empty bracket).
-268-9 explicitly state for what noise sources reference [52] refer to and also at approximately what sound levels.
-429 I would opt for the more usual definition of noise that is unwanted sound.

Reviewer 2 Report

After reading the submitted paper, I found myself a bit confused about the type of paper that it should be. The document start being a very good review to be shifted toward a suggested procedure. However, the suggested procedure is only described in bullet for a page length. This unbalance in the part confused me a lot. Please consider this to re-edit the text accordingly and decide if to produce a review paper, or expand significantly the result section, as well as the conclusions/discussion part that are nearly missing.

Further observations:

Please avoid the use of “we” in scientific writing, especially if the paper is a single author one.

I would also mention the possibility to use noise annoyance to derive noise limits, as reported in: Fredianelli, Luca, Stefano Carpita, and Gaetano Licitra. "A procedure for deriving wind turbine noise limits by taking into account annoyance." Science of the total environment 648 (2019): 728-736. Or Davy, John L., Kym Burgemeister, and David Hillman. "Wind turbine sound limits: Current status and recommendations based on mitigating noise annoyance." Applied acoustics 140 (2018): 288-295.

Reviewer 3 Report

The paper “A proposal to use determinants of annoyance in wind farm planning and management” investigates the objective and subjective evaluation of noise due to wind farm. It is a very interesting topic, but the paper can be considered quite similar to a review paper than an article.

The paper can be reorganized and exposed with more clarity. There is the lack of a methodology of reviewing and the outputs of the revision, on the planning and management of the wind farm, are only listed without the support of any application. The conclusions are very short compared with the entire paper.

It is suggested to pay attention to spelling and punctuation errors. Moreover, the text should be reorganized and written in a more understandable way. Finally, the reference to Wikipedia at line 344 should be removed and it is suggested to find a better reference for the definition of fear.

The paper cannot be accepted and published in the present form.

Reviewer 4 Report

Very interesting and actual paper considering the new problem of the energy transition.

The problem of accepting the construction of a wind farm is very complex.

In some cases there were javellin effect problems.

And a lot depends on the distance of the tower from the dwellings and the type of power.

There are also the undesirable effects from mini wind pover (less than 200 kW) which are louder than higher power towers.

Thanks for your attention.

Round 2

Reviewer 2 Report

The author fulfilled all my request and suggestions, and my judgment is now positive.

Author Response

Thank you for your positive response and your earlier constructive remarks.

As suggested, I will apply a final spell check

Reviewer 3 Report

After the first revision, the manuscript “A proposal to use determinants of annoyance in wind farm planning and management” has been improved and the author modified the text according to my requests. However, my opinion is that this manuscript should be published as a review paper. Indeed, my request of supporting the proposed approach with an application has been answered with the reference to a good practice already applied in the city of Amsterdam. So my question now is: where is the novelty of the approach if it has been already applied by others?

Moreover, another remark regards improving the writing skills from a formal point of view: the bulleted lists are not correct, but appear to be a mix between a list and simple lines of text. In addition, the text should be justified.

Finally, it is suggested to replace the word "chapter" in the introduction with "section" or "part".

Author Response

I can understand your wish for a proper review, because existing reviews dealing with social involvement or participation should be updated. I hope this paper will inspire some to think of such a review.

I did not mention it explicitly, but I was part of the small expert committee and their advice is very much based on my ideas. To be clear about this, I have added "including the author", where the expert committee is mentioned. 

Reviewer: However, my opinion is that this manuscript should be published as a review paper.

Author: The analysis in my paper shows that the lack of social acceptance is a major obstacle to further growth of onshore wind energy. It also shows that the resulting opposition is because residents will be exposed to 'objective' (visual and aural) factors, but also, and probably even more important, have 'subjective' concerns about health and fairness/respect. To make this clear, I had to show there is sufficient knowledge that supports my view on these two issues (objective/subjective). This background was necessary to explain the reason and substance of my recommendations. Reviewer 3 (and in the first round also reviewer 2) interpreted this background to be a review and recommended to change the paper so it would be a proper review. As I explained then, my paper is not and was not meant to be a (systematic) review: the recommendations are the essence of my paper, most of the rest of the text is to show why they are important. Apparently reviewer 2 accepted my explanation, reviewer 3 not. I can understand the wish and even need for an up to date scientific review of social acceptance of wind farms (not of wind energy in general), but this will need another paper. To publish my paper as a review paper, as reviewer 2 remarks, would require a quite different paper.

Reviewer: Indeed, my request of supporting the proposed approach with an application has been answered with the reference to a good practice already applied in the city of Amsterdam. So my question now is: where is the novelty of the approach if it has been already applied by others?

 Author: I found this comment/question difficult to understand. In the first round reviewer 3 remarked that the recommendations "are only listed without the support of any application". My answer was that the general lack of applying this type of approach was precisely the reason to write about it: the available scientific knowledge is not translated into practical measures. On the contrary: financial compensation is now a widely used measure, and although (as I remarked in the text) this certainly may be of help, it is known not to address the main reasons for opposition (and can even fuel opposition). In my revised text I added that my new approach has been applied in a recent report which included my input, so it was not "already applied by others". Does reviewer 3 mean that an approach that has been applied (by others or not) does not merit a scientific publication? 

Reviewer: Moreover, another remark regards improving the writing skills from a formal point of view: the bulleted lists are not correct, but appear to be a mix between a list and simple lines of text. In addition, the text should be justified.

Finally, it is suggested to replace the word "chapter" in the introduction with "section" or "part".

Author: I have followed up on the other suggestions as much as possible. It is true the bulleted list in 3.2 and the numbered list in 3.4 are different. In 3.2 is a list of conclusions, each part of the sentence above the list. in 3.4 are recommendations, each given as a separate sentence: nr. 2 was not, but is now corrected. 

Thank you for your observations.     

Round 3

Reviewer 3 Report

the paper can be accepted and published without any modification.

This manuscript is a resubmission of an earlier submission. The following is a list of the peer review reports and author responses from that submission.

Round 1

Reviewer 1 Report

Wind farm noise management based on determinants of annoyance is presented in the research. It seems that the article is written in hurry as many of the sections are incomplete.

The article requires formatting as per the journal format.

Abstract should be written in single paragraph followed by the significance of the research.

Introduction section is too short. Please provide suitable information of the complete article in the introduction section and the organization of the article at the end of the section.

Other sections also need special attention as they are too wordy without the support of the figures / tables etc., that are important in scientific research for better comprehension.

Conclusion is incomplete and should be re-written.

Reviewer 2 Report

I revised the manuscript entitled "Wind farm noise management based on determinants of annoyance", and my comments are below. 

  1. The author should follow the Journal's template.
  2. The references in the main text aren't presented correctly.
  3. The introduction section is very poor. It doesn't provide the current state of the research field and it doesn't cite almost any publications.
  4. Fig. 1 quality is very poor.
  5. The author includes reference from wikipedia, which in my oponion is no scientific content and it shouldn't be included in a research paper.
  6. The findings of this manuscript and the conclusions are very poor. The manuscript has nothing significant to mention. 

In my opinion, the present manuscript is below the journal standards and has no scientific interest.

I recommend that this manuscript should be rejected.

Reviewer 3 Report

The authors have attempted to design a Wind farm noise management based on determinants of annoyance.  However there are significant issues with this paper that have to be addressed before it can be accepted

1) the authors did not use the Journal format in preparing their manuscript.

2) What is the novelty of the work. I suggest the authors state the novelty in the introduction

3) No critical evaluation of the literature in the introduction.

4) No result and discussion section.

5) What is practical application of this work. I suggest the authors include this section